# Association of the Timeless Gene with Prognosis and Clinical Characteristics of Human Lung Cancer

**DOI:** 10.3390/diagnostics12112681

**Published:** 2022-11-03

**Authors:** Jishi Ye, Jingli Chen, Juan Wang, Zhongyuan Xia, Yifan Jia

**Affiliations:** 1Department of Pain, Renmin Hospital of Wuhan University, 99 Zhang Road, Wuhan 430060, China; 2Department of Anesthesiology, The Central Hospital of Wuhan, Tongji Medical College, Huazhong University of Science and Technology, Wuhan 430060, China

**Keywords:** timeless, lung cancer, *ESPL1*, bioinformatic analysis, prognosis

## Abstract

(1) Background: As the most common malignant tumor type worldwide, it is necessary to identify novel potential prognostic biomarkers to improve the poor prognosis of lung cancer. The Timeless gene, a circadian rhythm-related gene, is associated with several types of cancer. However, studies analyzing the clinical significance of the Timeless gene in patients with lung cancer are currently limited. (2) Methods: In the present study, the expression levels and prognostic potential of the Timeless gene and its co-expressed genes in different subtypes of lung cancer were explored using multiple bioinformatics approaches. The correlations between the Timeless gene and its co-expressed genes were validated using A549 and NCI-H226 cells by transfecting them with expression vectors and analyses using Western blot and reverse transcription-quantitative PCR. (3) Results: The Oncomine and GEPIA database analyses indicated that the expression of the Timeless gene was significantly higher in lung cancer as compared to that in the normal tissue. Using the UALCAN database, significant differences in Timeless gene expression were determined among different stages of lung cancer and between genders. A Kaplan–Meier plotter analysis indicated that high expression of the Timeless gene was associated with poor overall survival (OS) and progression-free survival (PFS) of patients with lung cancer. In the cBioPortal and GEPIA database analyses, extra spindle pole bodies like 1 (*ESPL1*) was the top correlated gene of Timeless in patients with lung cancer. Similar to the Timeless gene, high expression of the *ESPL1* gene was also associated with poor OS and PFS. Of note, overexpression of the Timeless gene increased the expression level of *ESPL1* at both the mRNA and protein levels. (4) Conclusion: The present study explored the clinical significance of the Timeless gene and its correlated gene *ESPL1* in patients with lung cancer, thereby providing a potential therapeutic target for lung cancer.

## 1. Introduction

Lung cancer is the most common malignant tumor type worldwide. Lung cancer ranked third among the most prevalent cancers and first among the most common causes of cancer-related death in Europe, with 388,000 deaths in 2018 alone [1]. The five-year survival of patients with lung cancer has also been low in the past decades. Despite the considerable progress in the diagnosis and treatment of cancers, the prognosis of lung cancer has not been improved, yet. Its poor prognosis urges researchers to identify the underlying molecular mechanisms of lung cancer, as well as the potential prognostic biomarkers and novel therapeutic strategies.

Timeless is a highly conserved protein-coding gene, which is involved in cell survival after damage or stress. This gene is also involved in the increase in DNA polymerase epsilon activity, and maintenance of telomere length and epithelial cell morphogenesis. It also has a role in the circadian auto-regulatory loop by interacting with the Period genes (*Per1*, *Per2* and *Per3*), and others to downregulate the activation of *Per1* via the CLOCK/ARNTL pathway [2]. Recent studies using cell lines, animal models and human tissues demonstrated that the Timeless genes serve as oncogenes, having roles in the development of numerous cancer types. For instance, knockdown of the Timeless gene was able to markedly inhibit the cellular proliferation rate in breast and cervical cancer cell lines [3]. In human colon cancer cell lines, the expression level of the Timeless gene was negatively correlated with the microRNA-139-5p level [4]. However, numerous studies on the role of the Timeless gene in the diagnosis and prognosis of patients with lung cancer are relatively superficial [5,6]. In the present study, the relationship between extra spindle pole bodies like 1 (*ESPL1*) and Timeless and their expression in lung cancer was further studied. Timeless may be useful as a diagnostic and prognostic marker for lung cancer and targeting it may be of high therapeutic value for this disease.

The present study analyzed the expression and clinical significance of Timeless and its co-expressed genes in the different subtypes of lung cancer using multiple biological databases. Subsequently, Timeless and its co-expressed genes were bioinformatically analyzed using Gene Ontology (GO) and the Kyoto Encyclopedia of Genes and Genomes (KEGG) pathways’ analyses. The present study may benefit researchers in deeply understanding the role of Timeless and its co-expressed genes in lung cancer.

## 2. Materials and Methods

### 2.1. Oncomine Database Analysis

The Oncomine database (https://www.oncomine.org/resource/login.html (accessed on 1 January 2020)) is a web-based data-mining platform containing cancer microarray data. This database was used to investigate the mRNA expression levels of the Timeless gene in lung cancer and adjacent normal tissues. The data were statistically compared. The cut-off criterion was set as *p* < 0.05 and fold-change (FC) > 2.

### 2.2. GEPIA Database

GEPIA (http://gepia.cancer-pku.cn/ (accessed on 1 January 2020)) is a newly developed interactive web server, which analyzes the RNA expression profiles of 9736 tumor samples and 8587 normal samples, extracted from The Cancer Genome Atlas (TCGA) and the GTEx projects using a standard processing pipeline. In the present study, only the median expression data of the Timeless gene in the tumor and normal samples were extracted from the GEPIA database and analyzed using the bodymap.

### 2.3. UALCAN Database

The UALCAN database (http://ualcan.path.uab.edu/ (accessed on 1 January 2020)) is also an interactive web portal for in-depth analyses of gene expression data from TCGA. This database was used to analyze the correlations between Timeless gene expression and clinicopathological parameters of patients with lung cancer, such as the cancer stage, age, gender and smoking habits of the patients.

### 2.4. Human Protein Atlas (HPA) Database

The HPA (https://www.proteinatlas.org/ (accessed on 1 January 2020)) is a multi-functional platform, which aims to map all the human proteins in cells, tissues and organs by integrating various omics technologies, such as antibody-based imaging, mass spectrometry-based proteomics, transcriptomics and systems biology. This database was used to search for the immunohistochemical (IHC) staining images of the Timeless gene in normal and lung cancer tissues.

### 2.5. Kaplan–Meier (K-M) Plotter

The K–M plotter (http://www.kmplot.com/ (accessed on 1 January 2020)) is capable of assessing the effects of any gene or gene combination on the survival of patients with breast, ovarian, lung, gastric, colon, prostate and 14 other types of cancer using >50,000 samples, which are measured using gene arrays, RNA-seq or next-generation sequencing technologies. In the present study, this tool was used to evaluate the prognostic potential of the Timeless gene in the different subtypes of lung cancer.

### 2.6. cBioPortal for Cancer Genomics

The cBioPortal (https://www.cbioportal.org/ (accessed on 1 January 2020)) for Cancer Genomics provides the visualization and analysis of large-scale cancer genomics datasets, which may be downloaded for manual analysis. In this study, the lung squamous cell carcinoma and lung adenocarcinoma datasets (TCGA, provisional) were selected to investigate the genetic alterations, correlations and network analysis of the Timeless gene in patients with lung cancer. The results provided the 25 most frequently altered neighbor genes, among which several genes were positively correlated with Timeless gene expression in lung squamous cell carcinoma and lung adenocarcinoma.

### 2.7. Database for Annotation, Visualization and Integrated Discovery (DAVID)

The DAVID v6.8 (https://david.ncifcrf.gov/ (accessed on 1 January 2020)) comprises a full knowledge base, updated to the sixth version of the original web-accessible programs. DAVID provides a comprehensive set of functional annotation tools for researchers to understand the biological annotations of large gene sets. GO and KEGG pathway enrichment analyses were performed using the 25 most frequently altered neighbor genes of Timeless in lung cancer.

### 2.8. Protein–Protein Interaction Analysis

The Search Tool for the Retrieval of Interacting Genes and proteins (STRING: https://cn.string-db.org/ (accessed on 1 January 2020)) is a database of known and predicted protein–protein interactions, including direct (physical) and indirect (functional) interactions. The interactions are predicted using computational algorithms based on the knowledge transfer between organisms and interactions extracted from other (primary) databases. This database was used to construct the protein–protein interaction network among the Timeless gene and the 25 most frequently altered neighbor genes in both squamous cell lung carcinoma and lung adenocarcinoma.

### 2.9. A549 and NCI-H226 Cell Culture and Transfection

The human lung adenocarcinoma and squamous carcinoma cell lines, A549 and NCI-H226 cells, respectively, were obtained from the Type Culture Collection of the Chinese Academy of Sciences. The A549 cells were cultured in DMEM (HyClone; Cytiva, Tokyo, Japan) supplemented with 10% FBS (HyClone; Cytiva) and penicillin (100 U/mL)/streptomycin (100 U/mL) and incubated at 37 °C in a humidified atmosphere with 5% CO_2_. The NCI-H226 cells were cultured in 10% RPMI 1640 medium (HyClone; Cytiva). The human TIMELESS cDNA ORF-cloned expression vector (cat. HG18793-UT) and empty control pCMV3-untagged vectors were obtained from SinoBiological (Beijing, China). The A549 and NCI-H226 cells were transfected with the expression or empty control vectors for 48 h using Lipofectamine^®^ 3000 (Thermo Fisher Scientific, Inc., Waltham, MA, USA).

### 2.10. Reverse Transcription-Quantitative (RT-q)PCR

The mRNA expression levels of the *ESPL1* gene in the two lung cancer cell lines (A549 and NCI-H226) were analyzed. For this purpose, the total cellular RNAs were extracted using TRIzol reagent (Invitrogen; Thermo Fisher Scientific, Inc.). The extracted RNA samples were reverse transcribed into cDNA using the iScript cDNA Synthesis kit (Bio-Rad Laboratories, Inc., Hercules, CA, USA) following the manufacturer’s instructions. Subsequently, the mRNA expression of the *ESPL1* gene was measured using qPCR with the following primer pair: forward, 5′-GCCCTAAAACTTACAACAAA-3′ and reverse, 5′-AGACTCAAGCAAGAACAGAA-3′). The RT-qPCR was performed using the iTaq Fast SYBR Green Supermix (Bio-Rad Laboratories, Inc.) under the following cycle conditions: 50 °C for 2 min, 95 °C for 20 s, (95 °C for 1 s, 60 °C for 20 s) × 40 repeats. The relative mRNA expression levels of the *ESPL1* gene were determined using the 2^−ΔΔCq^ method. All the experiments were performed independently in triplicate.

### 2.11. Western Blot Analysis

Cells were washed with PBS and harvested using the RIPA buffer (Beyotime, Nanjing, China). Protein concentration was determined by BCA assay (Beyotime, Nanjing). Total protein (20 µg) was separated using 12% SDS-PAGE, transferred to a nitrocellulose membrane and blocked with 5% milk at room temperature for 1 h. After denaturation, the total protein was separated using SDS-PAGE and then transferred onto polyvinylidene difluoride (PVDF) membranes (Beyotime, Nanjing). After blocking the proteins with 5% skimmed milk (Beyotime, Nanjing) for 1 h, the PVDF membranes were incubated with rabbit anti-mouse monoclonal antibodies against TIMELESS (1:1000 dilution; cat. no. 109512; Abcam), ESPL1 (1:500 dilution; cat. no. PAB0608; Abnova, Taipei, China) and GAPDH (1:1000 dilution; cat. no. 8245; Abcam, Cambridge, UK) overnight at 4 °C in a shaking incubator. The membranes were then washed with Tris-buffered saline containing Tween-20 and incubated with the respective anti-rabbit IgG secondary antibodies (1:2000 dilution; cat. no. 150077; Abcam) for 1.5 h at room temperature. The immunoblots were visualized using the Odyssey Infrared Imaging System (Li-COR Biosciences, Inc., Lincoln, NE, USA).

### 2.12. Statistical Analysis

All of the data were presented as the mean ± standard deviation. Statistical analyses were performed using GraphPad Prism 5 (GraphPad Software Inc., San Diego, CA, USA). The differences between the experimental and control groups were analyzed for statistical significance by Student’s *t*-test. *p* < 0.05 was considered to indicate statistical significance. The statistical results and methods of public databases are derived from the statistical software built into these databases and were not deliberately processed.

## 3. Results

### 3.1. Expression Level of the Timeless Gene in Human Lung Cancer

The expression levels of the Timeless gene were displayed using the Oncomine and GEPIA databases. The mRNA expression levels of the Timeless gene in various cancer types using pooled analyses are presented in Figure 1A. The results revealed that the Timeless gene was overexpressed in breast, brain, cervical, colorectal, head and neck and lung cancer tissues as compared to their matched normal tissues. The median expression of the Timeless gene in the tumor and normal tissue samples was analyzed using the bodymap from the GEPIA database (Figure 1B). The Oncomine database also revealed that the expression levels of the Timeless gene were significantly higher in small cell lung carcinoma [7], squamous cell lung carcinoma [7,8,9], large cell carcinoma [8] and lung adenocarcinoma [8,10] (Figure 2). Furthermore, IHC staining images, acquired from the HPA database, also confirmed that the expression levels of the Timeless gene were upregulated in squamous cell lung carcinoma and lung adenocarcinoma as compared to the matched healthy lung tissues, which further validated the results from the Oncomine database (Figure 3). In the HPA database, we found that Timeless was localized in the cytoplasm. Furthermore, the expression of the Timeless gene in healthy lung tissues was from a sample of a female patient aged 67 years (A; patient ID, 2208). The expression of the Timeless gene in lung adenocarcinoma was from a sample of a male patient aged 67 years (B; patient ID, 4886), and the expression of Timeless in lung squamous cell carcinoma was from a sample of a female patient aged 61 years (C; patient ID, 4900). 

### 3.2. Expression of Timeless Gene and Clinicopathological Parameters of Patients with Lung Cancer

The UALCAN database was used to analyze the associations between Timeless gene expression and clinicopathological parameters of patients with lung cancer, including lung squamous cell lung carcinoma and lung adenocarcinoma. The associations of Timeless gene expression with lung squamous cell carcinoma were analyzed. The results revealed higher expression levels of the Timeless gene in patients with cancer stages 1, 2 and 3 as compared to those in patients with cancer stage 4 and normal tissues (*p* < 0.05; Figure 4). The gender-specific gene expression analysis suggested that the expression of the Timeless gene was significantly upregulated in male and female patients with lung squamous cell lung carcinoma as compared to the normal group (*p* < 0.05). For lung adenocarcinoma, the analysis indicated that the expression of the Timeless gene was significantly upregulated in male patients as compared to female patients and the normal group (*p* < 0.05). However, there were no significant differences among different age groups or ethnicities.

The prognostic potential of the Timeless gene was analyzed using the K–M plotter (Figure 5). Among the patients with lung squamous cell carcinoma and lung adenocarcinoma, a high expression of the Timeless gene was associated with poor overall survival (OS) and progression-free survival (PFS) (Figure 5A–D); however, in patients with lung squamous cell carcinoma, there was no significance (Figure 5E,F). For both male and female patients, the increased Timeless gene was also associated with unfavorable OS. The Timeless gene was of prognostic value for patients with lung cancer, irrespective of their smoking habits. 

### 3.3. Genetic Mutations, Correlations and Networks Analysis of the Timeless Gene in Patients with Lung Cancer

The genetic mutations, correlations and network analyses of the Timeless gene in lung squamous cell carcinoma and lung adenocarcinoma (TCGA, provisional) were performed using the cBioPortal tool. The results indicated that the Timeless gene was mutated in 10 out of 230 (4%) patients with lung adenocarcinoma and 3 out of 178 (2%) patients with lung squamous cell carcinoma. The co-expression analysis of the Timeless gene in the patients with lung squamous cell lung carcinoma and lung adenocarcinoma was performed using the cBioPortal tool and the GEPIA database (Figure 6 and Figure 7). Both for lung squamous cell lung carcinoma and lung adenocarcinoma, *ESPL1* and Timeless were highly co-expressed with a strong correlation. *ESPL1* was the top correlated gene, which was also confirmed using the GEPIA database.

In order to confirm the correlations between *ESPL1* and the Timeless gene, A549 and NCI-H226 cells were transfected with Timeless expression vector (Figure 8A,B). Overexpression of the Timeless gene significantly increased the expression of *ESPL1* at both the mRNA and protein levels. 

Furthermore, a protein–protein interaction network for the Timeless gene and the 25 most frequently altered neighbor genes in both lung squamous cell lung carcinoma and lung adenocarcinoma was constructed by using the STRING and DAVID databases (Figure 8C). As presented in Figure 9, a GO enrichment analysis indicated that the Timeless gene and its altered neighboring genes were mainly enriched in DNA replication, regulation of signal transduction by p53 class mediator and DNA damage checkpoint in the category biological process. In the category cellular component, these genes were mainly enriched in the DNA replication factor C complex and Ctf18 RFC-like complex in the nucleoplasm. Furthermore, these genes were mainly enriched in DNA clamp loader activity, single-stranded DNA-dependent ATPase activity and DNA binding in the category molecular function. The KEGG enrichment analysis for the Timeless gene and its frequently altered neighboring genes indicated that the mismatch repair, circadian rhythm and DNA replication may be involved in the oncogenic pathogenesis of lung cancer.

### 3.4. ESPL1 Expression and Prognostic Potential in Patients with Lung Cancer

Based on the analysis with the Oncomine database (Figure 10), *ESPL1* expression was identified to be significantly overexpressed in the five lung cancer datasets, including squamous cell lung carcinoma [7,8] (FC = 2.741 and 2.447, respectively), large cell carcinoma [8] (FC = 4.807) and lung adenocarcinoma [11,12] (FC = 2.249 and 2.339, respectively).

The prognostic potential of *ESPL1* was similar to that of the Timeless gene in lung cancer (Figure 11). Among the patients with lung squamous cell carcinoma and lung adenocarcinoma, high expression of *ESPL1* was associated with poor OS and PFS [*p* = 4.1 × 10^−12^, hazard ratio (HR) = 1.57 (95% CI, 1.38–1.78) and *p* = 2.1 × 10^−9^, HR = 1.78 (95% CI, 1.47–2.15), respectively]. High expression of *ESPL1* mRNA was also significantly associated with poor OS only in patients with lung adenocarcinoma [*p* = 1.2 × 10^−5^, HR = 1.69 (95% CI, 1.34–2.15)]. However, there was no significant association between high expression of *ESPL1* mRNA and OS in patients with lung squamous cell carcinoma [*p* = 0.7, HR = 1.05 (95% CI, 0.82–1.35)].

## 4. Discussion

The present study comprehensively analyzed the mRNA expression levels of the Timeless gene and its most significantly correlated gene *ESPL1* in different subtypes of human lung cancer using multiple databases. First, a critical role of Timeless in the expression profile and prognosis of lung cancer was indicated. The associations of Timeless gene expression with clinicopathological parameters, including the cancer stage, gender, prognosis, race and age of patients with lung cancer, were also presented in this study. The GO and KEGG enrichment analyses identified the functional terms and pathways related to the Timeless gene and the frequently altered neighbor genes. These signaling pathways, including mismatch repair, circadian rhythm and DNA replication, have been demonstrated to have important roles in the oncogenic mechanisms of lung cancer. For instance, among the patients with non-small cell lung cancer, the genetic polymorphisms in the mismatch repair pathway may be potential clinical markers for the prediction of chemotherapeutic toxicity [13]. Furthermore, disruption of the circadian rhythm was also reported to be a potential risk factor for cancer development and poor prognosis, indicating the inhibitory effect of circadian rhythm homeostasis on the tumor [14]. 

The Timeless gene has an important role in the control of DNA replication, maintenance of replication fork stability, maintenance of genome stability throughout normal DNA replication and regulation of the circadian clock [15,16,17,18,19]. Given its potential role in the determination of period length and DNA damage-dependent phase advancing of the circadian clock, numerous studies have explored the correlations between the Timeless gene and tumorigenesis [20,21,22,23,24]. Chi et al. [21] demonstrated that the expression levels of the Timeless gene in both the breast cancer tissues and cell lines increased significantly. Knockdown of the Timeless gene was able to reduce the tumorigenicity of breast cancer cells in vivo, which may involve the well-known oncogene *MYC* [21]. These results were consistent with those of the previous studies on breast cancer [25,26,27]. This may provide a basis for using this gene as a therapeutic target for breast cancer. Although numerous studies indicated that the Timeless gene was overexpressed in cervical cancer, studies focusing on the underlying mechanism of this gene’s role in cervical cancer are limited. Zhang et al. [24] suggested that the overexpression of the Timeless gene was associated with pelvic lymph node metastasis, demonstrating the role of the lympho-vascular space in human cervical cancer. This gene may have the potential of being used as an independent predictive biomarker for poor OS and PFS of patients with early-stage cervical cancer. However, its underlying molecular mechanisms are required to be explored further [24]. Similarly, studies focusing on the underlying mechanisms of the Timeless gene in lung cancer are also limited. A single-center study reported that the Timeless gene was overexpressed in surgically resected specimens among 88 consecutive patients. High protein levels of Timeless were associated with poor OS. Cytological experiments also confirmed that knockdown of this gene inhibited the growth of lung cancer cells and induced apoptosis in H157 and H460 cells [28]. Combining these results with those of the present study, it was demonstrated that the Timeless gene may be a valuable diagnostic and prognostic biomarker for lung cancer. Therefore, this gene should be focused on to explore its high therapeutic potential for lung cancer.

*ESPL1* is a protein-coding gene and initiates the final separation of sister chromatids, which is a critical step for chromosomal inheritance [29]. Based on the GTEx database, *ESPL1* is overexpressed in the esophagus-mucosa, vagina and testis. Current studies indicated that *ESPL1* may also participate in the formation and progression of the tumor. For instance, as a cell cycle-associated gene, the overexpression of *ESPL1* was significantly associated with a favorable prognosis. Furthermore, the expression level of *ESPL1* was negatively associated with the pathologic stage/progression of gastric adenocarcinoma. However, *ESPL1*/separase has been recognized as a putative oncogene of luminal B breast cancers [30]. In luminal breast cancers, the increased expression of *ESPL1* was associated with poor prognosis of patients of the luminal B subtype. Animal experiments also verified that overexpression of *ESPL1* in luminal tumors may be associated with the loss of tumor suppressor genes (*P53* and *Rb*). In addition to lung cancer, studies also indicated that *ESPL1* and five other genes were significantly upregulated in small cell lung carcinoma as compared to normal cells, lung squamous cell carcinoma, lung adenocarcinoma and large cell carcinoma. Furthermore, studies also suggested that overexpression of *ESPL1* and the other five genes negatively affected OS and relapse-free survival [31]. Therefore, these genes may be used as powerful prognostic biomarkers and potential targets for patients with lung cancer. In order to confirm the association between the *ESPL1* and Timeless genes, A549 and NCI-H226 cells were transfected with Timeless gene expression vectors. Overexpression of the Timeless gene significantly increased the expression level of *ESPL1* at both the mRNA and protein levels. A549 and NCI-H226 cells represented lung squamous cell carcinoma and lung adenocarcinoma, respectively. The cell culture and transfection experiments reflected the association between the *ESPL1* and Timeless genes, which preliminarily confirmed the significant role of these genes in the development of tumors. However, the regulatory regions and downstream regulatory elements are still required to be further studied. Altogether, these results suggested that the Timeless gene and its co-expressed *ESPL1* gene may have great prognostic and treatment potential in lung cancer.

However, there are certain limitations to this study. First, the present study only explored the expression levels and prognostic potential of the Timeless and *ESPL1* genes in lung cancer using several public databases. However, this study did not validate these findings using PCR or IHC. Furthermore, although the GO and KEGG enrichment analyses were able to screen out certain related pathways and biological processes, limited studies have explored the underlying mechanisms of the effects of Timeless on the prognosis of patients with cancer. Considering the limited funds available, the expressions of Timeless and ESPL1 were only determined in lung cancer cell lines. In future experiments, the effect of Timeless on the invasion and migration in lung cancer cells and normal cells from patients may be investigated to further confirm the clinical significance of Timeless in lung cancer.

In conclusion, the present study comprehensively analyzed the mRNA expression levels of the Timeless gene and its most significantly correlated gene *ESPL1* in different types of human lung cancer and predicted the underlying mechanisms of these genes in lung cancer, thereby providing a better understanding of lung cancer pathogenesis. These results suggested that high expression of the Timeless and *ESPL1* genes in lung cancer tissues may have an important role in the development of lung cancer. Therefore, these genes may serve as potential prognostic biomarkers for the improvement of survival of patients with lung cancer and prognostic accuracy.

## Figures and Tables

**Figure 1 diagnostics-12-02681-f001:**
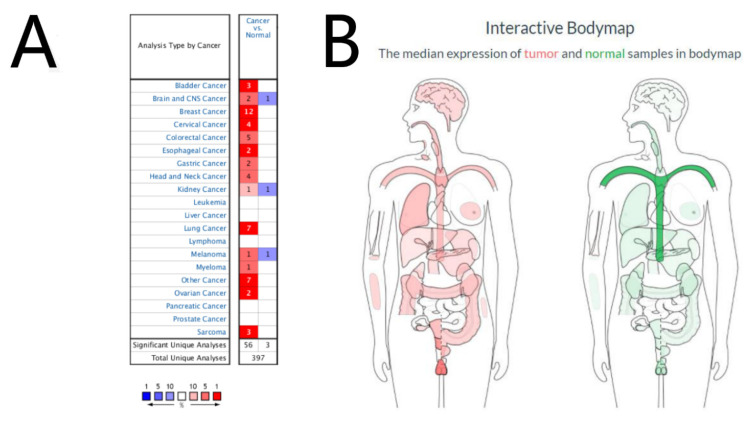
Expression profile of the Timeless gene in different human tissues. (**A**) Expression of Timeless in different types of cancer from the Oncomine database. (**B**) Median expression of tumor and normal samples in the bodymap from the GEPIA database.

**Figure 2 diagnostics-12-02681-f002:**
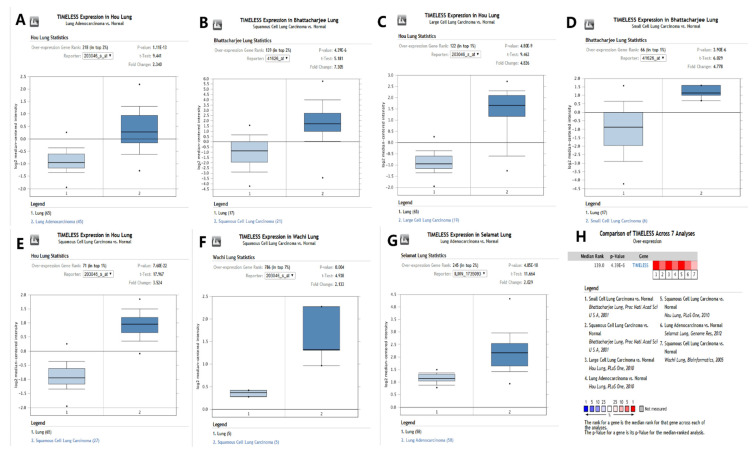
Expression profile of the Timeless gene from (**A**–**G**) Oncomine database. Expression profile of Timeless gene in seven lung cancer datasets. (**H**) Comparison of Timeless gene in the seven analyses.

**Figure 3 diagnostics-12-02681-f003:**
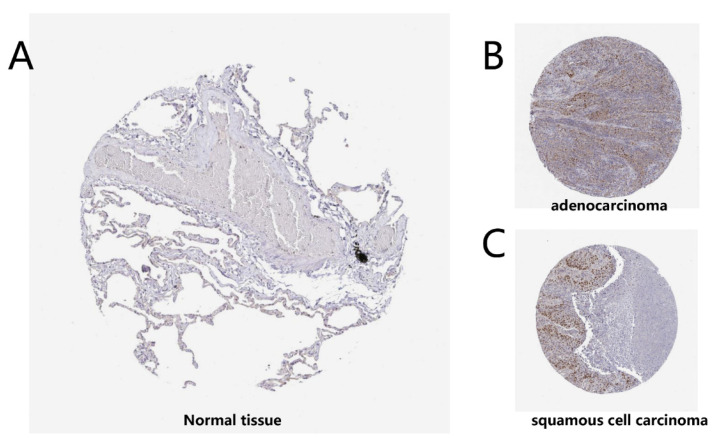
Immunohistochemical analysis of the Timeless gene in the Human Protein Atlas database. (**A**) Expression of the Timeless gene in healthy lung tissues (female; 67 years; patient ID, 2208). (**B**,**C**) Expression of the Timeless genes in lung adenocarcinoma (male; 67 years; patient ID, 4886) and lung squamous cell carcinoma (female; 61 years; patient ID, 4900).

**Figure 4 diagnostics-12-02681-f004:**
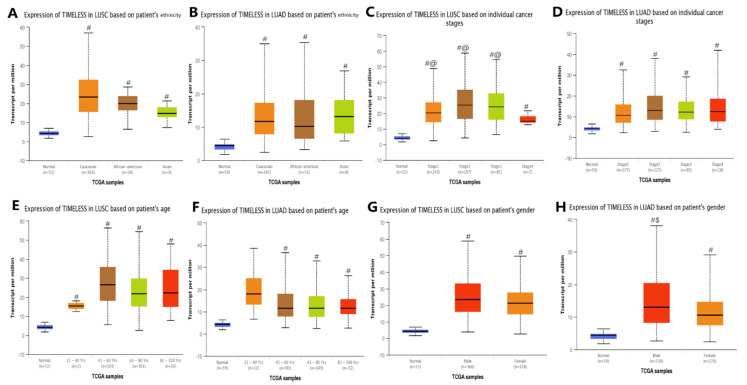
Association of Timeless gene expression with clinicopathological parameters of patients with lung cancer. (**A**,**C**,**E**,**G**) Timeless gene expression and clinicopathological parameters of patients with LUSC. (**B**,**D**,**F**,**H**) Timeless gene expression and clinicopathological parameters of patients with LUAD. ^#^ *p* < 0.05 vs. normal group; ^@^ *p* < 0.05 vs. cancer stage 4; ^$^ *p* < 0.05 vs. female group. TCGA, The Cancer Genome Atlas; LUSC, lung squamous cell carcinoma; LUAD, lung adenocarcinoma.

**Figure 5 diagnostics-12-02681-f005:**
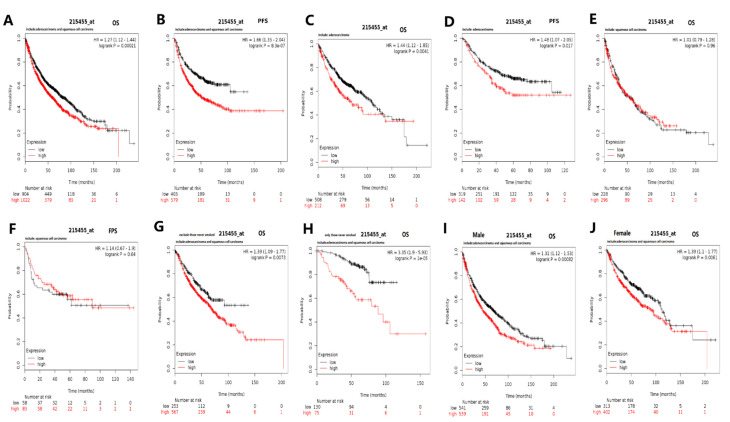
Prognostic potential of the Timeless gene in patients with lung cancer analyzed using Kaplan–Meier plotter. High expression of the Timeless gene was associated with (**A**,**C**) poor OS and (**B**,**D**) poor PFS of patients with lung squamous cell carcinoma and lung adenocarcinoma. (**G**,**H**) Timeless gene expression was of prognostic value for patients with lung cancer irrespective of their smoking habits. (**I**,**J**) High expression of the Timeless gene was associated with poor OS of both males and females. (**E**,**F**) Timeless gene expression was associated with poor OS and PFS of patients with lung squamous cell carcinoma, while it was not associated with those of patients with lung squamous cell carcinoma. PFS, progression-free survival; OS, overall survival; HR, hazard ratio (presented with 95% CI).

**Figure 6 diagnostics-12-02681-f006:**
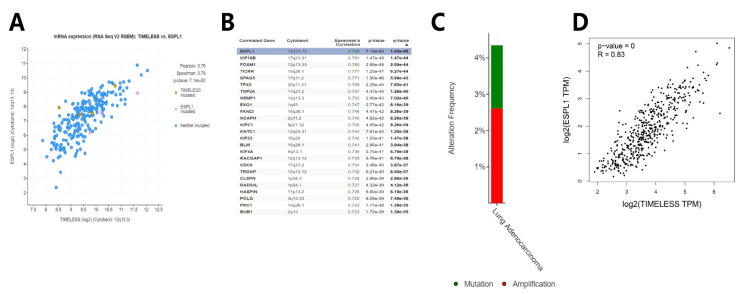
Genetic mutational and correlational analysis of Timeless gene in the patients with lung adenocarcinoma. (**A**,**B**) Correlation analysis between the Timeless gene and its co-expressed genes using the cBioPortal tool. (**C**) Genetic mutational analysis of the Timeless gene in lung adenocarcinoma. (**D**) Pearson correlation analysis between Timeless and *ESPL1* genes using the GEPIA database. ESPL1, extra spindle pole bodies like 1.

**Figure 7 diagnostics-12-02681-f007:**
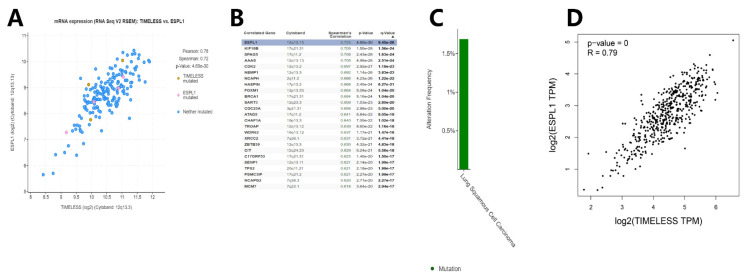
Genetic mutational and correlational analysis of Timeless genes in the patients with lung squamous cell carcinoma. (**A**,**B**) Correlation analysis between the Timeless gene and its co-expressed genes using the cBioPortal tool. (**C**) Genetic mutational analysis of the Timeless gene in lung squamous cell carcinoma. (**D**) Pearson correlation analysis between Timeless and *ESPL1* genes using the GEPIA database. ESPL1, extra spindle pole bodies like 1.

**Figure 8 diagnostics-12-02681-f008:**
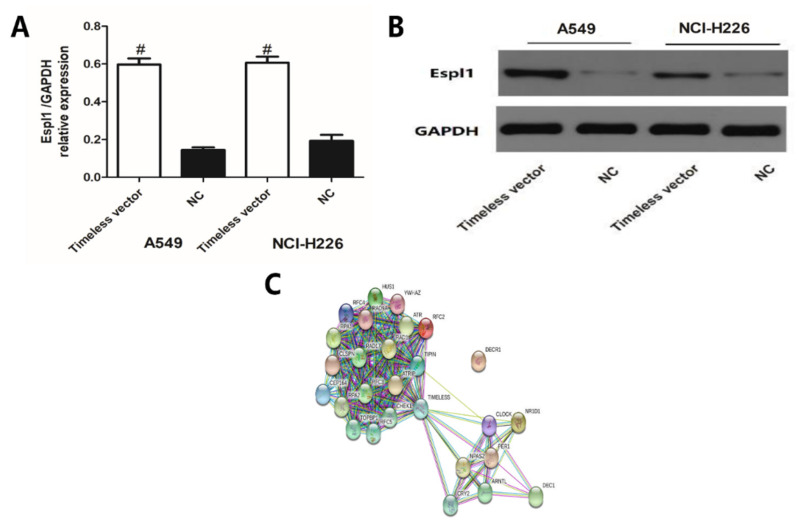
(**A**,**B**) *ESPL1* expression at mRNA and protein levels in the A549 and NCI-H226 cells after 48 h of transfection with the Timeless expression vector or the NC vector. (**C**) Protein–protein interaction network for the Timeless gene and the 25 most frequently altered neighbor genes in both squamous cell lung carcinoma and lung adenocarcinoma using the STRING database. NC, negative control; ESPL1, extra spindle pole bodies like 1. ^#^ *p* < 0.05 vs. negative control.

**Figure 9 diagnostics-12-02681-f009:**
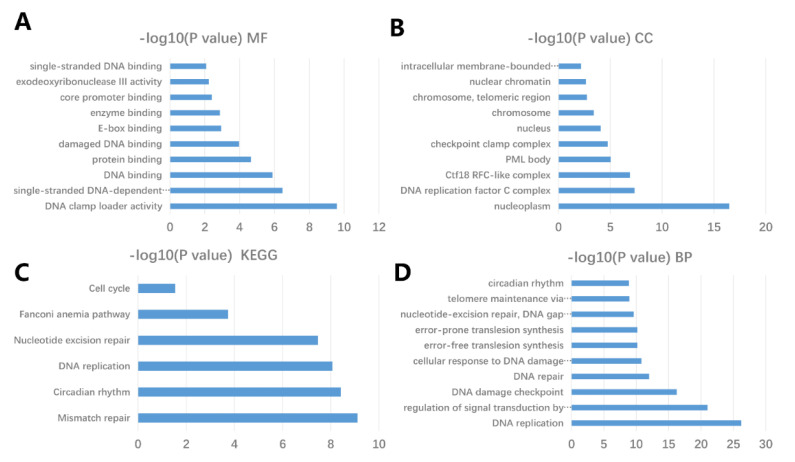
GO and KEGG enrichment analyses for the Timeless gene and the 25 most frequently altered neighbor genes in lung cancer performed using the DAVID database. (**D**) GO analysis suggested that the Timeless gene and its altered neighbor genes were mainly enriched in BP terms including DNA replication, regulation of signal transduction by p53 class mediator and DNA damage checkpoint. (**B**) GO analysis in the category CC indicated that these genes were mainly enriched in DNA replication factor C complex and Ctf18 RFC-like complex in the nucleoplasm. (**A**) GO analysis in the category MF suggested that the genes were mainly enriched in DNA clamp loader activity, single-stranded DNA-dependent ATPase activity and DNA binding. (**C**) KEGG pathway enrichment analysis of the genes indicated that mismatch repair, circadian rhythm and DNA replication may be involved in the oncogenesis and mechanism of lung cancer. GO, Gene Ontology; MF, molecular function; CC, cellular component; BP, biological process; KEGG, Kyoto Encyclopedia of Genes and Genomes.

**Figure 10 diagnostics-12-02681-f010:**
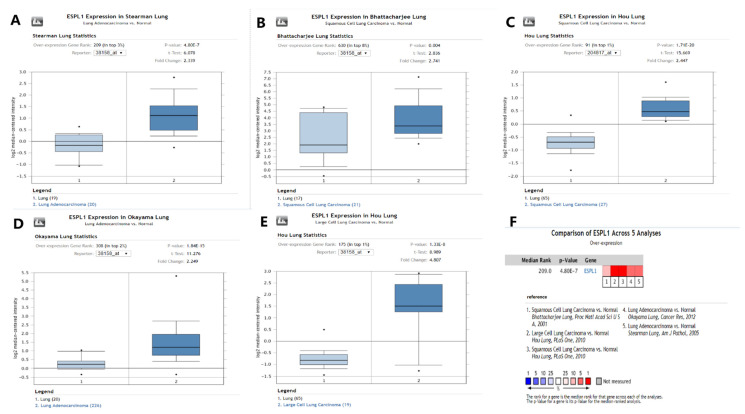
Expression profile for *ESPL1* from the Oncomine database. (**A**–**E**) Expression of Timeless gene in the five lung cancer datasets. (**F**) Comparison of *ESPL1* across the five datasets. ESPL1, extra spindle pole bodies like 1.

**Figure 11 diagnostics-12-02681-f011:**
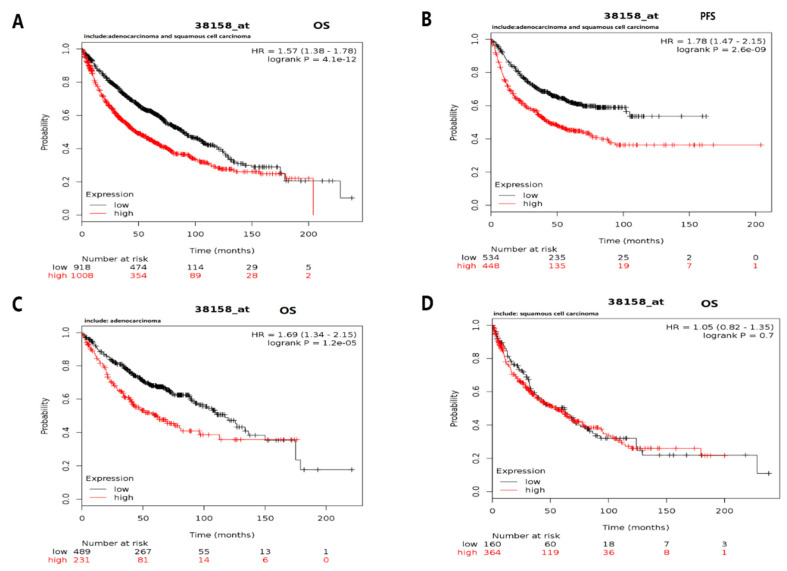
Prognostic potential of *ESPL1* in patients with lung cancer determined using Kaplan–Meier plotter. For patients with lung squamous cell carcinoma and lung adenocarcinoma, high expression of the *ESPL1* gene was associated with (**A**) poor overall survival and (**C**) poor PFS. mRNA overexpression of *ESPL1* was significantly associated with poor OS only in (**B**) patients with lung adenocarcinoma, but (**D**) not in patients with lung squamous cell carcinoma. PFS, progression-free survival; OS, overall survival; HR, hazard ratio (presented with 95% CI); ESPL1, extra spindle pole bodies like 1.

## Data Availability

The datasets analyzed during the current study are available from the corresponding author upon reasonable request.

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
