# Peer review of "Association of the Timeless Gene with Prognosis and Clinical Characteristics of Human Lung Cancer"

_diagnostics, 2022, doi:10.3390/diagnostics12112681_

Round 1
Reviewer 1 Report
Dear Editor,
I have reviewed the work of Ye, et al., titled: “Association of the Timeless gene with prognosis and clinical 2 characteristics of human lung cancer”. This study is a investigation paper on the role of Timeless gene in both culture cells and lung cancer. In general, The work attempts to explain the role of the Timeless gene and its associations in lung carcinoma. To carry out this study, different genomic platforms were used, which demonstrated the expression of the gene in lung cancer compared to normal tissue, in addition to demonstrating ESPL1 with prognostic factors in patients with this cancer.
The study could be interesting for the journal, however there are different aspects that must be modified before the work could be considered for publication. Here are my comments:
Point 1: The study involves patients with lung cancer, which implies that the Institutional Review Board Statement is necessary.
Point 2: The study involves patients with lung cancer, why was the informed consent of the patients avoided?
Point 3: How many patients with lung carcinoma were analyzed in the study? What were the demographic characteristics of the study group? What were the inclusion and exclusion criteria? It is important to know the clinical stage of the patients, as well as the treatment received.
Point 4: The authors note that the Timeless gene was demonstrated in different tissues. They should establish the reason for using different tissues, and not focus on lung tissue. I don't understand.
Point 5: How was the antibody obtained for the demonstration of the Timeless gene in tissue? Was the tissue used fresh or from paraffin-embedded material? How was the optimal dilution of the antibody determined? Was any antigen retrieval methodology used? What was the positive control? The entire methodology in materials and methods must be developed.
Point 6: The discussion should take into account the results of the study. However, the authors discuss the findings in other types of cancer, so it is necessary to rewrite.
Author Response
Point 1: The study involves patients with lung cancer, which implies that the Institutional Review Board Statement is necessary.
Response:Thank you for your question. In this study, we did not use any tissue from patients. All data came from public database. The immunohistochemical (IHC) staining images of the Timeless gene in normal and lung cancer tissues are also from HPA database, which was a public and multi-functional platform, which aims to map all the human proteins in cells, tissues and organs by integrating various omics technologies, such as antibody-based imaging, mass spectrometry-based proteomics, transcriptomics and systems biology. Given these, we think that it is not necessary to get the Ethics approval and consent.
Point 2: The study involves patients with lung cancer, why was the informed consent of the patients avoided?
Response:Thank you for your question. Although this study involved patients with lung cancer, we did not any tissue and data in our hospital’s patients. All data were from the public bioinformatic databases, such as Oncomine, GEPIA, UALCAN, Kaplan-Meier (K-M) plotter and cBioPortal for Cancer Genomics. These database own vast amounts of information and data about cancer patients. All people can freely use and analyze these data. So it is not need to sign a patient consent form.
Point 3: How many patients with lung carcinoma were analyzed in the study? What were the demographic characteristics of the study group? What were the inclusion and exclusion criteria? It is important to know the clinical stage of the patients, as well as the treatment received.
Response:Thank you for your question. Although this study involved patients with lung cancer, we did not any tissue and data in our hospital’s patients. All data were from the public bioinformatic databases.
GEPIA is a newly developed interactive web server for analyzing the RNA sequencing expression data of 9,736 tumors and 8,587 normal samples from the TCGA and the GTEx projects, using a standard processing pipeline。
Point 4: The authors note that the Timeless gene was demonstrated in different tissues. They should establish the reason for using different tissues, and not focus on lung tissue. I don't understand.
Response:thank you for your question. In this study, we just verified the expression of Timeless gene and its co-expressed gene in A549 and NCI-H226 cells by transfecting them using western blot and reverse transcription-quantitative PCR. Consideration the limited funds, we cannot demonstrate the expression of Timeless in different tissues. Some papers also focused the clinical significance of Timeless in different tissues and diseases.
Point 5: How was the antibody obtained for the demonstration of the Timeless gene in tissue? Was the tissue used fresh or from paraffin-embedded material? How was the optimal dilution of the antibody determined? Was any antigen retrieval methodology used? What was the positive control? The entire methodology in materials and methods must be developed.
Response:thank you for your question. Are the antibody concentrations you refer to the IHC antibody concentrations? The immunohistochemical (IHC) staining images of the Timeless gene in normal and lung cancer tissues are also from HPA database, which was a public and multi-functional platform. In this database, you can find the immunohistochemical staining images of the Timeless gene in patients’ tissues and their ID, sex and years. But this database did not mention the entire methodology in materials and methods.
Point 6: The discussion should take into account the results of the study. However, the authors discuss the findings in other types of cancer, so it is necessary to rewrite.
Response:thank you for your question. In the first paragraph of the discussion section, we mentioned the results of this study. The present study comprehensively analyzed the mRNA expression levels of the Timeless gene and its most significantly correlated gene ESPL1 in different subtypes of human lung cancer using multiple databases. First, a critical role of Timeless in the expression profile and prognosis of lung cancer was indicated. The associations of Timeless gene expression with clinicopathological parameters, including the cancer stage, gender, prognosis, race and age of patients with lung cancer, were also presented in this study. The GO and KEGG enrichment analyses identified the functional terms and pathways related to the Timeless gene and the frequently altered neighbor genes. These signaling pathways, including mismatch repair, circadian rhythm and DNA replication, have been demonstrated to have important roles in the oncogenic mechanisms of lung cancer. For instance, among the patients with non-small cell lung cancer, the genetic polymorphisms in the mismatch repair pathway may be potential clinical markers for the prediction of chemotherapeutic toxicity. Furthermore, disruption of the circadian rhythm was also reported to be a potential risk factor for cancer development and poor prognosis, indicating the inhibitory effect of circadian rhythm homeostasis on the tumor.
Reviewer 2 Report
The authors have developed an extremely interesting study examining the association of the Timeless gene with prognosis and clinical features of lung cancer. The manuscript is very well-documented and holds great interest for readers. However, there are a few problems I have highlighted, as well as some suggestions I have for authors:
ABSTRACT: The summary is too long. There is also no need for sub-headings in it. Please provide an informative and balanced summary in the abstract.
INTRODUCTION:
-The introduction explains very well the scientific background and justification of the investigation that is the subject of the report. However please state specific objectives, including any prespecified hypotheses.
METHODS:
-Please present key elements of study design.
-Please describe all statistical methods
DISCUSSIONS:
-It would also be interesting to mention (and I recommend that you do so) other markers that have clinical relevance in lung cancer, for example p63 and TTF-1 (e.g: https://pubmed.ncbi.nlm.nih.gov/31263838/ ; https://pubmed.ncbi.nlm.nih.gov/12760288/)
Author Response
ABSTRACT: The summary is too long. There is also no need for sub-headings in it. Please provide an informative and balanced summary in the abstract.
Response 1:thank you for your advise. We had amended the abstract. Please check it.
INTRODUCTION:
-The introduction explains very well the scientific background and justification of the investigation that is the subject of the report. However please state specific objectives, including any prespecified hypotheses. Our hypotheses is that
Response:thank you for your advise. But we don’t know what is the prespecified hyptheses.
Our hypotheses is that the Timeless gene and its correlated gene ESPL1 is related with the diagnosis and prognosis of patients with lung cancer, thereby providing a potential therapeutic target for lung cancer.
In introduction, we stated that main meaning of this study. The present study analyzed the expression and clinical significance of Timeless and its co-expressed genes in the different subtypes of lung cancer using multiple biological databases. The present study may benefit researchers in deeply understanding the role of Timeless and its co-expressed genes in lung cancer.
METHODS:
-Please present key elements of study design.
Response:thank you for your advise.
Overall, we used several public database to look for the correlationship between Timeless gene and the diagnosis and prognosis of patients with lung cancer. By cell experiments, we verified the expression of the Timeless gene and its correlated gene ESPL1 in human lung adenocarcinoma and squamous carcinoma cell lines. Of course, In future experiments, the effect of Timeless on invasion and migration in lung cancer cells and normal cells from patients may be investigated to further confirm the clinical significance of Timeless in Lung cancer.
-Please describe all statistical methods
Response:thank you for your advise.
All of the data were presented as the mean ± standard deviation. Statistical analyses were performed using GraphPad Prism 5 (GraphPad Software Inc.). The differences between the experimental and control groups were analyzed for statistical significance by Student’s t-test. P<0.05 was considered to indicate statistical significance. The statistical results and methods of public databases are derived from the statistical software built into these databases and were not deliberately processed.
DISCUSSIONS:
-It would also be interesting to mention (and I recommend that you do so) other markers that have clinical relevance in lung cancer, for example p63 and TTF-1 (e.g: https://pubmed.ncbi.nlm.nih.gov/31263838/ ; https://pubmed.ncbi.nlm.nih.gov/12760288/)
Response : thank you for your advise.
p63 and TTF-1 were useful tumor markers for lung cancer. The p63 positivity and TTF-1 negative expression consequently indicated a poorly differentiated nonkeratinizing SCC, while the opposite immunostaining pattern was flagged in SCLC. However, we found that p63 and TTF-1 had no strongly correlationship with Timeless in lung cancer, which no much aritcles on it can be retrieved. So we cannot mention these two articles in our studies. Please forgive us. Thank you.
Round 2
Reviewer 1 Report
The authors have adequately responded to the requests made.